# Impact of Commercial Seaweed Liquid Extract (TAM^®^) Biostimulant and Its Bioactive Molecules on Growth and Antioxidant Activities of Hot Pepper (*Capsicum annuum*)

**DOI:** 10.3390/plants10061045

**Published:** 2021-05-21

**Authors:** Mohamed Ashour, Shimaa M. Hassan, Mostafa E. Elshobary, Gamal A. G. Ammar, Ahmed Gaber, Walaa F. Alsanie, Abdallah Tageldein Mansour, Rania El-Shenody

**Affiliations:** 1National Institute of Oceanography and Fisheries, NIOF, Cairo 11516, Egypt; 2Department of Vegetable Crops, Faculty of Agriculture (El-Shatby), Alexandria University, Alexandria 21545, Egypt; shaymaa.hassan@alexu.edu.eg; 3Botany Department, Faculty of Science, Tanta University, Tanta 31527, Egypt; rania.elshnody@yahoo.com; 4Biotechnology Unit, Plant Production Department, Arid Lands Cultivation Research Institute, City of Scientific Research and Technological Applications, Alexandria 21934, Egypt; gammar@srtacity.sci.eg; 5Department of Biology, College of Science, Taif University, P.O. Box 11099, Taif 21944, Saudi Arabia; a.gaber@tu.edu.sa; 6Department of Clinical Laboratories Sciences, The Faculty of Applied Medical Sciences, Taif University, P.O. Box 11099, Taif 21944, Saudi Arabia; w.alsanie@tu.edu.sa; 7Animal and Fish Production Department, College of Agricultural and Food Sciences, King Faisal University, P.O. Box 420, Al-Ahsa 31982, Saudi Arabia; amansour@kfu.edu.sa; 8Fish and Animal Production Department, Faculty of Agriculture (Saba Basha), Alexandria University, Alexandria 21531, Egypt

**Keywords:** hot pepper, clean production, antioxidants, TAM^®^, seaweed biostimulants, phytochemicals

## Abstract

Bioactive molecules derived from seaweed extracts are revolutionary bio-stimulants used to enhance plant growth and increase yield production. This study evaluated the effectiveness of a commercially available seaweed liquid extract, namely, True-Algae-Max (TAM^®^), as a plant growth stimulant on nutritional, and antioxidant activity of *Capsicum annuum*. Three concentrations of TAM^®^ (0.25, 0.5, and 1%) of various NPK: TAM^®^ ratios were investigated via foliar spray, over 2017 and 2018 cultivation seasons, under greenhouse conditions. TAM^®^ is rich in phytochemical compounds, such as ascorbic acid (1.66 mg g^−1^), phenolics (101.67 mg g^−1^), and flavonoids (2.60 mg g^−1^) that showed good antioxidant activity (54.52 mg g^−1^) and DPPH inhibition of 70.33%. Promoting measured parameter results stated the extensive potentiality of TAM^®^ application, in comparison with conventional NPK treatment. Yield and composition of *C. annuum* were significantly improved in all TAM^®^ treated groups, especially the TAM_0.5%_ concentration, which resulted in maximum yield (4.23 Kg m^−2^) and significant amounts of profuse biological molecules like chlorophyll, ascorbic, phenolic compounds, flavonoids, and total nutrients. Compared to the NPK control treatments, *C. annuum* treated with TAM_0.5%_ improved the total antioxidant activity of hot Pepper from 162.16 to 190.95 mg g^−1^. These findings indicate that the extract of seaweed can be used as an environmentally friendly, multi-functional biostimulant in the agricultural field for more sustainable production, in addition to reducing the use of hazardous synthetic fertilizers.

## 1. Introduction

The utilization of traditional mineral fertilizers has been increasing dramatically due to the rapid growth in the global population, and constantly augmented food demand [1]. While synthetic fertilizers can increase crop yields and development, their widespread use has serious negative consequences, such as soil salinity and hardening, which can lead to decreased soil fertility, pesticide reinforcement, and water pollution [2]. Therefore, there is a strong demand for developing and operating novel alternative inputs in crop production. The concept of biostimulants application in agriculture has been explored since 1933 but has attracted interest in recent times as an eventual means to alleviate the negative effects of climate change on the agricultural sector and enhance plant growth and defense mechanisms towards various stresses as well [3]. The effective application is the use of biostimulants in the form of mixtures of phytochemicals that could induce plant growth and yield while they improve protection against biotic and abiotic stresses, without any drawbacks [4,5,6,7]. Marine organisms are rich in phytochemical biomolecules with diverse biological activities [8,9,10,11,12,13,14]. In this context, seaweeds would have been wielded as biostimulants [5,6,15,16].

Overall, seaweeds are an untapped source of several organic and inorganic components; improving plant quantity and quality, by promoting plant growth, besides defense and immune response [17,18,19]. The Egyptian coastline, including the Mediterranean [20,21,22], and Red Sea coasts [16,19,23], has a heterogeneous variation of wild seaweed genera that could regenerate and been collected throughout the year. The most common native seaweed species along Alexandria’s Mediterranean coast, Egypt are red alga; *Pterocladia capillacea, Jania rubens*, and green alga; *Ulva lactuca*. The biochemical, mineral composition and antioxidant activity of these organisms have all been well studied previously [24] as well as antimicrobial activity [21,24,25]. Seaweed extracts have been demonstrated for their plant defense, aiding roles against several environmental stresses due to harsh conditions (drought, salinity, high or low temperature, dearth of essential elements), or other diseases caused by biotic factors [26]. Induction to plants resistance is usually related to the overproduction of pathogen-inhibited proteins and metabolites [27,28,29]. These variations in biochemical composition can result in beneficial changes in the crop’s nutritional and pro-health quality.

Bioactive molecules derived from seaweed extracts are revolutionary biostimulants used to enhance plant growth and increase yield production. The commercial seaweed liquid extract, namely; True-Algae-Max (TAM^®^), is a natural plant growth regulator that contains high phytochemical compounds, such as polysaccharides, nutrients, and many other bio-molecules [30], which showed good promoting measured parameters on plant growth and increase yield production of rocket salad (*Eruca vesicaria*) [6] and cucumber (*Cucumis sativus*) [7]. Interestingly, 5-Silaspiro[4.4]nona-1,3,6,8-tetraene, 3,8-bis (diethylboryl)-2,7-diethyl-1,4,6,9-tetraphenyl-, Milbemycin-oxime, and Nonadecane, were novel phytochemical compounds were reported in TAM^®^ [6,7,30].

Genus *Capsicum* is a part of the family *Solanaceae*, which includes several high-value vegetables such as; tomatoes, potatoes, and eggplants among more than 90 genera and 2500 species of flowering plants [31]. This genus is native to tropical and subtropical America in a wide area, including; Mexico and northern Central America [32], the Caribbean, the lowlands of Bolivia, the northern lowlands of Amazonia, and the Southern Andes mid-elevation, where archeological evidence indicates that this category of crops has been used since 6000 BC [33,34]. Hot Pepper *Capsicum*
*annuum* is an important universal vegetable crop in terms of its commercial value and rural economic significance [35,36]. On the one hand, biostimulants represent precious biostimulants to increase plant growth, yield, and quality in sustainable farming systems, especially the decreased practice of industrial synthetic fertilizers to potentially minimize its harmful impact on the agricultural environment for better future sustainability [37]. Since seaweed extracts and their physiological effects have been well documented, the influence of such extracts on phytochemicals and the enhancement of their activities has been understudied and requires further research. Therefore, the current study aims to determine the effect of TAM^®^ as a foliar spray on vegetative growth, yield, and changes in the nutraceutical and nutritional quality of Hot Pepper (*Capsicum annuum*). In this context, the performance of seaweed biofertilizers (three treatments with 50% chemical fertilizer) and absolute fertilizer was compared, in terms of vegetative growth, yield, nutrient content, and bioactive compounds.

## 2. Materials and Methods

### 2.1. Seaweeds

Seaweed extract growth regulator, formally named “True-Algae-Max” (TAM^®^), is a research-output commercial product (*know-how*), submitted as a patent [38]. Preparing of TAM^®^ was previously described by Ashour et al. [30]. TAM^®^ has a seaweed odor and its color is dark-brown, with pH and density of 9–9.5 and 1.2, respectively. TAM^®^ biochemical composition showed total organic matter of 23.2% (based on dry matter, DM), total polysaccharides of 15% DM, total dissolved solids of 2.6% DM. Hence, major nutrients values were 12%, 2.4%, and 1400 mg kg^−1^ for total potassium, phosphorus, and nitrogen, respectively. Microelements of copper (0.39 mg kg^−1^), iron (16.18 mg kg^−1^), magnesium (19.72 mg kg^−1^), zinc (1.19 mg kg^−1^), and manganese (3.72 mg kg^−1^) were reported. Heavy metal analysis showed that cadmium, chromium, lead, and nickel were less than the limit of quantification, as reported by Ashour et al. [30]. Moreover, phytochemical analysis of crude TAM^®^ was analyzed using GC-Mass spectrophotometry analysis as described Elshobary et al. [12], however, phytochemical compounds of TAM^®^ were previously investigated by Ashour et al. [30] as cited in Table 1.

### 2.2. Hot Pepper (Capsicum annuum) Methods

#### 2.2.1. Soil

During the 2017 and 2018 seasons, the current trial was carried out under greenhouse conditions (300 m^−2^) at Abis Experimental Farm Station (31.2001° N and 29.9187° E), Faculty of Agriculture, Alexandria University, Egypt. Before starting the experiment, soil samples were collected at 15–30 cm depth, and analyzed at the Central Laboratory, Faculty of Agriculture, Alexandria University. According to Evenhuis et al. [45], the physical and chemical properties of soil were determined, as seen in Table 2.

During the growing trials, the treatments of crude TAM^®^ (TAM_0.25%_, TAM_0.5%_, and TAM_1%_) were added twice weekly as a foliar spray (100–200 mL m^−2^), while the control NPK treatment (TAM_0%_) was applied, in equivalent doses, once weekly via a drip irrigation system, starting one week after transplanting. After transplanting, nitrogen fertilizers in the form of calcium nitrate (17% N), phosphorus fertilizers in the form of phosphoric acid (61.5% P_2_O_5_), and potassium fertilizers in the form of potassium sulphate (48% K_2_O) with a total amount of 12.6 kg, 1.9 L 100 m^−2^, and 8 kg 12 weeks, respectively. The schedule of fertigation system was in accordance with the recommendations of the Ministry of Agriculture and Land Reclamation of Egypt for the commercial production of Hot Pepper *C. annuum* (Omega F1 cv.) [31], as presented in Table 3.

#### 2.2.2. Experimental Design

Seeds of *C. annuum* were transplanted in plastic houses on the 1st of February in both seasons (2017 and 2018).The current experimental trial was conducted to investigate the impact of 50% partial replacement of traditional NPK mineral fertilization by three different levels (2.5 ppt, 5 ppt and 10 ppt) of seaweed foliar spray TAM^® (^TAM_0.25%_, TAM_0.5%_ and TAM_1%_, respectively), as a biostimulant on growth, yield, fruit characterizations (yield, fruit length, and diameter), and antioxidant activities (ascorbic acid, total flavonoids compounds, total phenolic compounds, total antioxidant capacity, and DPPH inhibition%) of a colored marketable stage of Hot Pepper; *Capsicum annuum* (Omega F1 cv.), cultured under greenhouse conditions. In the current experiment, four foliar spray levels of crude TAM^®^ (TAM_0%_, TAM_0.25%_, TAM_0.5%_ and TAM_1%_) were applied; TAM_0%_ (0 mL L^−1^) of crude TAM^®^, as a control 100% NPK classical chemical fertilizer; TAM_0.25%_ (2.5 mL L^−1^), TAM_0.5%_ (5 mL L^−1^), and TAM_1%_ (10 mL L^−1^) of crude TAM^®^ using of only 50% of the recommended NPK classical chemical fertilizer in terms of chemical fertilizer reduction. Three replicates were applied for each level (treatment). For each replicate, four plots (each plot is 1 m wide and 20 m long with a total area of 20 m^2^) were conducted. The experimental layout was a Randomized Complete Blocks Design (RCBD). Seeds of *C. annuum* were transplanted in plastic houses on the 1st of February in both seasons (2017 and 2018). The average maximum and minimum temperatures during the four growing months from February to May were 33.5 and 13.5 °C, respectively. Average maximum and minimum relative humidity was 79.8 and 73.5%, respectively.

### 2.3. Tested Parameters

#### 2.3.1. Hot Pepper *Capsicum annuum* Growth Parameters

At the end of the experiment, plant height (cm) and number of branches (n) were investigated. Random samples of five *C. annuum*, from each plot, were taken for determining leaves chlorophyll (mg 100 g fresh weight), dry matter (%), nitrogen (N), phosphorus (P), and potassium (K) contents (mg 100 g dry weight). Total chlorophyll content (µE = µMol m^−2^ s^−1^) of fresh leaves was calculate by using chlorophyll fluorometer (model OPTI-SCIENCES OS-30), Opti-sciences, Inc. NH, USA. Impurities were removed from each plot’s samples, which were then washed in purified water and oven-dried at 70 °C until they reached a steady weight. Then, the percentages of each leaf dry matter contents (%) were calculated. Total nitrogen, total phosphorus, and total potassium contents were figured out based on the dry matter of leaves. Total nitrogen and total phosphorus contents were checked calorimetrically, using a spectrophotometer at OD_662_ and OD_650_ nm [46], while total potassium was determined by atomic absorption spectrometry [47].

#### 2.3.2. *Capsicum annuum* Yield and Fruit Parameters

In each plot, pepper fruits, at a colored marketable stage, were harvested twice weekly. The total yield as an average to both 2017 and 2018 seasons was calculated by collected-picked fruit yield (Kg m^−2^) from each plot. The average fruit length (L) in cm was determined after measuring the length of ten fruits at random. Furthermore, the average head diameter in cm was determined after measuring fruit diameter (D) with a “venire caliper” for ten fruits selected at random.

#### 2.3.3. Antioxidant Activities of TAM^®^ and *Capsicum annuum* Fruits

Ascorbic acid (mg g^−1^) content, total flavonoid compounds (mg g^−1^), total phenolic compounds (mg g^−1^), total antioxidant capacity (mg g^−1^), and DPPH inhibition (%) of TAM^®^, in addition to a colored marketable stage of fruits of *C. annuum* were determined, based on dry matter bases, as described by El-Shenody et al. [19].

### 2.4. Statistical Analysis

The analysis of variance (ANOVA) was used to identify differences among means, that were considered significant at *p* < 0.05 degree of probability, getting the least significant difference (LSD) to resolve the differences among means of replication according to Duncan by SPSS for windows, statistical analysis software package; version 16.0.

## 3. Results

### 3.1. Antioxidant Activities of Crude TAM^®^

Crude TAM^®^ extract was tested for antioxidant activities (Figure 1). Total phenolic compounds showed the highest content of 101.67 mg g^−1^, while total flavonoid compounds, and ascorbic acid showed very little content of 2.60 and 1.66 mg g^−1^, respectively. These bioactive compounds revealed a good total antioxidant capacity of 54.52 mg g^−1^, and DPPH inhibition percentage of 70.33%, as presented in Figure 1.

### 3.2. Hot Pepper C. annuum

#### 3.2.1. Growth

Table 4 shows the effect of different levels of crude TAM^®^ on *C. annuum* growth characterization, cultured under greenhouse conditions, during 2017 and 2018 seasons. In general, comparing with the control group (TAM_0%_), *C. annuum* treated with TAM^®^ levels (TAM_0.25%_, TAM_0.5%_, and TAM_1%_) had the higher significant plant height, branches, chlorophyll, dry matter, leaf-P, and leaf-K, while achieving the lower significant leaf-N content (Table 4). The growth data revealed that compared to the control group (TAM_0%_), the height of plants treated with TAM_0.5%_, reported increasing rates of 30.70% and 17.76%, in both 2017 and 2018 seasons, respectively. Additionally, the number of branches has been significantly affected by applying TAM_0.25%_ and TAM_0.5%_, in both 2017 and 2018, with increasing rates of 28.48% and 38.57%, respectively, for each level.

TAM_0.5%_ level has significantly increased chlorophyll content with rates of 9.11% and 15.19% in both 2017 and 2018 seasons, respectively, more than the control group (TAM_0%_), followed by TAM_0.25%_ and TAM_1%_, correspondingly. Regarding the dry matter, TAM_0.5%_ improved the dry matter with rates of 10.94% and 9.70%, in both 2017 and 2018, consequently. For nutrients’ content, Table 4 figured out that, comparing to TAM_0_, TAM_0.25%_ and TAM_0.5%_ were able to significantly increase leaf-P content, with rates of 29.27% and 23.81% for TAM_0.25%_ and 25.20% and 20.64 for TAM_0.5%_ in 2017 and 2018, respectively. The same trend was observed for leaf-K content, where treatments of TAM_0.25%_ and TAM_0.5%_ displayed a significant increase over TAM_0%_ (Table 4). On the other hand, TAM_0_ displayed the highest leaf-N content followed, by TAM_0.25%_ and TAM_0.5%_, respectively, while TAM_1%_ showed the lowest leaf-N content with a decrease of 13.09% and 9.13% in 2017–2018, respectively (Table 4).

#### 3.2.2. Yield and Chemical Composition

Generally, during 2017 and 2018 repetitive seasons, applying different TAM^®^ levels (TAM_0.25%_, TAM_0.5%_, and TAM_1%_) could significantly increase total yield, fruit length, and fruit diameter of *C. annuum,* comparing to the control group (TAM_0%_), (Table 5). It was recorded that the total yield feature was significantly increased by TAM_0.5%_ with rates of 32.22% and 23.20%, over the standard control in 2017 and 2018, respectively, followed by TAM_0.25%_ and TAM_1%_, correspondingly. Likewise, approximately, the same pattern could be observed by results obtained from measuring fruit lengths and fruit diameter (Table 5).

#### 3.2.3. Antioxidant Activities of Colored Marketable Stage of Fruits of *C. annuum*

Predominantly, *C. annuum* treated with TAM^®^ levels (TAM_0.25%_, TAM_0.5%_, and TAM_1%_) had higher significance in Ascorbic acid, flavonoid, and phenolic content, as well as total antioxidant capacity, and DPPH inhibition percentage, when being compared to TAM_0%_. Among all applied TAM^®^ levels, TAM_0.5%_ had the highest significant ascorbic acid content (4.16 ± 0.15 mg g^−1^), total flavonoid compounds (3.47 ± 0.12 mg g^−1^), total phenolic compounds (113.63 ± 0.38 mg g^−1^), along with total antioxidant capacity (190.95 ± 4.18 mg g^−1^), and DPPH inhibition percentage (82.71 ± 2.09 mg g^−1^), achieving increasing rates of 52.85%, 32.79%, 48.79%, 17.75%, and 12.63%, higher than those of TAM_0%_, representing the control treated with 100% NPK traditional chemical fertilizers (Figure 2).

## 4. Discussion

Seaweeds extract bioregulator has several advantages over chemical fertilizers. They are characterized by their biodegradability, and innocuousness, turning them into environmentally friendly substances, with zero toxic residuals and/or threats [3]. Seaweed extracts and derivatives have significant biotechnological potentials for disease resistance, due to their unique and exceptional existence and biomolecular structure [5,6,7]. Aside from that, they have been committed to supplying nutrients for crops, promoting the production of superior biomass, and triggering plants’ natural ability to cope with environmental stresses. They are known to have a beneficial impact on plants due to their ability to produce several biologically active plant immune inciting molecules. Accordingly, seaweed extracts are strongly recommended to assist organic agriculture due to their very promising stable measures [3,5,6,7]. Native aquatic species should be tested for their industrial and biotechnological potential as a source of bioactive compounds [48,49,50,51,52,53].

The current study aimed to compare the effect of seaweed commercialized extract (TAM^®^) as a foliar application on pepper *C. annuum* vegetative growth, yield, and quality compared to the conventional NPK fertilization. The hypothesis tested, for this reason, was that foliar application of TAM^®^ can improve pepper agronomic traits, and fruit quality, on the physical and/or chemical basis. The application of different TAM^®^ concentrations effectively affected growth parameters in a very promising magnitude, according to the obtained results (Table 4). It seems that plant height, branches’ number, dry matter, and chlorophyll content, would display a significant difference over the two years of cultivation. In contrast, the effect of used doses across N, P, and K measurements, did not demonstrate the same influence among the control and TAM^®^ treatments. Table 5 revealed that treatments with different TAM^®^ concentrations could improve fruit quantity and quality parameters, such as total yield, length, and diameter of fruit compared to the NPK fertilizer, which also reflected on dry matter percentage, as well as for minerals contents. This findings may be attributed to the presence of several essential nutrients and bioactive compounds in seaweeds extract such as essential micro and macronutrients, vitamins, phytohormones, organic acids, and a variety of other bioactive components [5,6,17]. In addition, Arthur et al. [54] found that using seaweed extract could increase pepper yield.

According to Ashour et al. [30] who reported that seaweed extracts (TAM^®^) is a good growth promoter, a biochemical enhancer, and increase yield production of Rocket Salad *Eruca sativa* [6] and cucumber (*Cucumis sativus*) [7], as well as, interestingly, a growth and immunity enhancer of Nile Tilapia *Oreochromis niloticus*, challenged with *Aeromonas hydrophila* [30,55]. The current study found that crude TAM^®^ extract showed high antioxidant activities of total phenolic compounds, as well as, showed total flavonoid compounds and ascorbic acid. These bioactive compounds revealed a good total antioxidant capacity of 54.52 mg g^−1^, and DPPH inhibition percentage of 70.33% (Figure 1). It was known that the existed phenolics, flavonoids and ascorbic acid, as well as polysaccharides previously reported in TAM^®^, have positive effects as a growth promoter, a biochemical enhancer, increase yield production [6,7], and enhance growth and feed utilization, serum biochemistry, immune activity, and resistance to diseases, for several aquatic species [30]. Interestingly, 5-Silaspiro[4.4]nona-1,3,6,8-tetraene, 3,8-bis (diethylboryl)-2,7-diethyl-1,4,6,9-tetraphenyl-, Milbemycin-oxime, and Nonadecane, were novel phytochemical compounds were reported in TAM^®^ [6,7,30,55].

Recently, seaweed extracts used as a foliar spray have attracted considerable attention due to their ability to promote rapid growth and yield of cereals, fruits, and orchards, as well as horticultural plants [6]. TAM^®^ application can also vary from the control treatment in other ways. Regardless of its variable capacity to perform a direct significant difference with all examined calculated parameters, this is perceived as a benefit rather than being a flaw. Having the same impact with 100% traditional mineral fertilizers could open up the possibility of using seaweed extracts as biostimulants alternatively. If it is possible to fully or partially replace the use of chemical fertilizers, this would be a significant addition that would enable us to save our universe and atmosphere.

Enhancement of vegetative growth, yield, and bioactive ingredients of *C. annuum* were also reported earlier in the literature by different authors. Marhoon and Abbas [56], investigated the impact of seaweed extract at concentrations of 0, 3, or 6 ml L^−1^ on some vegetative, and anatomical parameters of the sweet pepper plant (*C. annuum* L.) and *California wonder*. The results revealed that seaweed extract at 6 ml L^−1^ showed a remarked increase in plant height, number of branches, and the percentage of dry matter of shoots. Furthermore, the thicknesses of the cortex and vascular cylinder, as well as the diameter of vascular units, were significantly increased when treated with the higher concentration of seaweed extract. Eris et al. [57] detected the effects of three different concentrations of seaweed extract (*Ascophyllum nodosum*) administered as a foliar spray at five different stages of pepper growth on yield and quality parameters (cv. *California wonder*). All treatment caused an increase in fruit yield, at 10-days earlier to the first harvest, compared with control. When increasing seaweed extract, fruit length and diameter increased significantly. Nonetheless, the difference in fruit wall thickness between treatments was not important. The amount of soluble solids and chlorophyll in the water has increased.

Seaweed extract treatment enhanced plant chlorophyll content when used as a foliar spray in a dose of 0.5% *Ascophyllum* extract at 10-day intervals [58]. This treatment also resulted in a significant enhancement of sweet pepper growth parameters, including plant height by 40%, leaf number by 50%, plant dry biomass by 52%, root length by 59%, and chlorophyll content by 20% over the controlled ones [58]. Similar results were observed in the present study on *C. annum*, where the highest chlorophyll content was recorded by TAM_0.5%_. Interestingly, *C. annum* treated with TAM_0.5%_ significantly presented higher amounts of ascorbic acid, total phenol, and total flavonoid compounds, compared to those obtained from the regular NPK control. Increasing the content of such compounds could have a positive effect on stirring up the immune response against biotic and abiotic stresses of the treated plant. Ali et al. [58] reported that applying *A. nodosum* extract could remarkably minimize disease incidence by pathogens, in tomato and sweet pepper, cultivated either under greenhouse, or field conditions, by suppressing disease mechanisms via inducing the activities of defense-related enzymes, and the content of total phenolic compounds. Yıldıztekın et al. [59] observed that the treatment by seaweed extract of *A. nodosum*, increased vegetative growth in *Capsicum annuum* L. at all concentrations applied under salinity conditions. Not only that but also maximizing the plant antioxidant enzyme activity under salt stress, when treated with the seaweed extract. From the obtained data, it is detected that foliar application of seaweeds extracts, displayed similar or higher values to those achieved in the control treatment. These outcomes are guiding recommendations to use and exploit seaweed extracts to maximize advantages from all their beneficial applications.

## 5. Conclusions

While it has a great potentiality to be an innovative environmentally friendly substitute for routinely applied NPK fertilizers, this research work has validated that the foliar spray of a commercial seaweed extract (True Algae Max, TAM^®^, Egypt) could seriously enhance the morpho-agronomic and bioactive properties of hot pepper, *C. annum*, comparatively with those attained through standard control treatment of regular chemical fertilizers. Enthusiastically, the biomolecules found in TAM^®^, were able to provide superior options for treated plants. Among the three practiced TAM^®^ levels, TAM_0.5%_ treatment resulted in the most effective treatment on almost all growth parameters, minerals, bioactive compounds, and antioxidant activities of *C. annum*. This study shows that TAM^®^ may be a reasonable tool for improving the morpho-agronomic, and bioactive traits of pepper plants. Furthermore, the product’s preparation is easy and has no side effects. These findings tightly match the worldwide governmental policies to save money, time, and efforts, in order to fulfill better lifestyles, health, and welfare, with sufficient economic and environmental benefits.

## 6. Patents

Seaweed extract (TrueAlgaeMax, TAM^®^) is a patent submitted at Egyptian Patents office, Academy of scientific research and technology (submission No.: 2046/2019).

## Figures and Tables

**Figure 1 plants-10-01045-f001:**
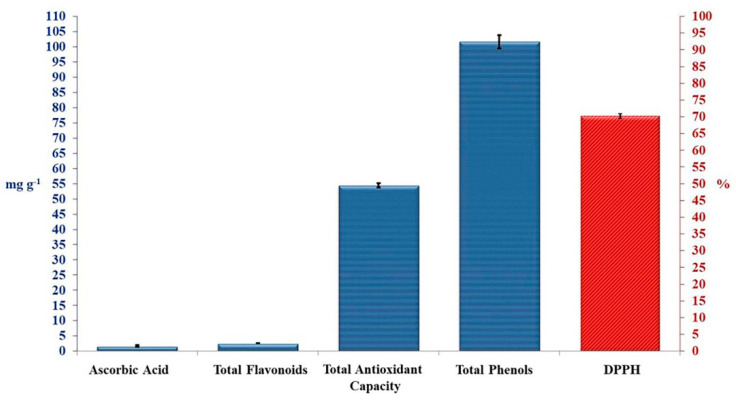
Antioxidant activities of crude TAM^®^. Error bars indicate standard deviations of three replicates.

**Figure 2 plants-10-01045-f002:**
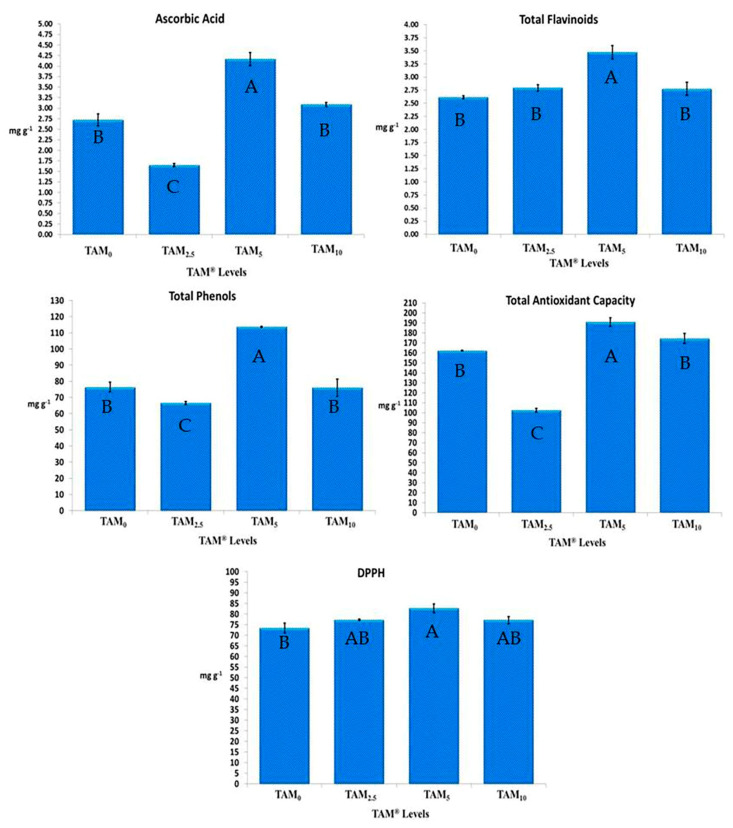
Antioxidant Activities of C. annuum treated with Crude TAM^®^. Error bars indicate standard deviations of three replicates. Different superscript letters in each row indicate significant differences (*p* < 0.05).

**Table 1 plants-10-01045-t001:** Phytochemical constituents of seaweed biostimulant, TAM^®^, with the most important biological activities according to literature *.

RT (min)	Phytochemical Compounds Name	Chemical Formula	Molecular Weight	Nature	Biological Properties	Literatures
8.99	5-Silaspiro[4.4]nona-1,3,6,8-tetraene,3,8-bis(diet-hylboryl)-2,7-diethyl-1,4,6,9-tetraphenyl-	C_44_H_50_B_2_Si	628.39	Silicon-boron compound	Fish and plant growth regulator and immunity enhancer	[6,7,30]
16.31	Nonadecane	C_19_H_40_	268.31	Alkane	Fish and plant immunity enhancer; antioxidant; antimicrobial; anti-inflammatory,	[6,7,30]
19.45	Rhodopin	C_40_H_58_O	554.45	Carotenoid	Fish and plant growth enhancer; antioxidant	[6,7,30,39]
20.07	Milbemycin-oxime	C_32_H_44_ClNO_7_	589.28	Macrocyclic lactones	Fish and plant immunity enhancer; antiparasitic; antihelmintic; insecticidal	[6,7,30]
20.90	Tridecanoic acid methyl ester	C_14_H_28_O_2_	228.21	Fatty acid methyl esters (FAMEs)	Antioxidant; herbicidal; antimicrobial; surfactants	[6,7,30,40]
21.63	Oleic Acid	C_18_H_34_O_2_	282.26	Fatty acid	Fish and plant immunity enhancer; anti-inflammatory	[6,7,30,40,41]
23.74	γ-Linolenic acid methyl ester	C_19_H_32_O_2_	292.24	FAMEs	Antioxidant; herbicidal; antimicrobial; surfactants	[6,7,30,41,42]
24.02	9,12-Octadecadienoic acid methyl ester, (E,E)-	C_19_H_34_O_2_	294.26	FAMEs	Antioxidant; herbicidal; antimicrobial; surfactants	[6,7,30,42,43]
24.37	Phytol	C_20_H_40_O	296.31	Diterpene alcohol	Antioxidant; plant growth enhancer	[30,39,44]

* Cited from Ashour et al. [30].

**Table 2 plants-10-01045-t002:** Chemical and physical characteristics of the soil at the experimental location, during the growing seasons of 2017 and 2018.

Soil Parameters	2017	2018
Particle size distribution		
Sand (%)	42.5 ± 3.6	39.8 ± 3.1
Silt (%)	25.2 ± 2.5	28.7 ± 2.2
Clay (%)	32.3 ± 1.6	31.5 ± 1.2
Soil texture	Sandy loam	Sandy loam
pH	7.45 ± 0.5	7.35 ± 0.3
Chemical Characteristics		
Soluble Cations (mmol g^−1^ soil)		
Ca^2+^	1.44 ± 0.4	1.40 ± 0.5
Mg^2+^	1.45 ± 0.4	0.98 ± 0.2
Na^+^	3.63 ± 0.5	4.75 ± 0.7
K^+^	0.54 ± 0.05	0.36 ± 0.03
Soluble Anions (mmol L^−1^)		
HCO_3−_	1.66 ± 0.1	1.78 ± 0.2
Cl^−^	2.00 ± 0.3	1.80 ± 0.2
SO_4_^2−^	1.70 ± 0.5	1.65 ± 0.6
Total nitrogen (TN) (%)	0.16 ± 0.03	0.15 ± 0.01
Available phosphorus (mg L^−1^)	0.32 ± 0.02	0.27 ± 0.01

**Table 3 plants-10-01045-t003:** Schedule of fertigation system (100% mineral fertilizer as a control treatment).

NPK Rate	Week after Transplanting	Calcium Nitrate (g 100 m^−2^)	Potassium Sulphate (g 100 m^2^)	Phosphoric Acid (cm^3^ 100 m^2^)
2%	2	252	160	38
4%	3	504	320	76
6%	4	756	480	114
8%	5	1008	640	152
12%	6	1512	960	228
12%	7	1512	960	228
12%	8	1512	960	228
12%	9	1512	960	228
8%	10	1008	640	152
8%	11	1008	640	152
8%	12	1008	640	152
8%	13	1008	640	152

**Table 4 plants-10-01045-t004:** The effect of different TAM^®^ levels on growth characterization of *C. annuum*, during 2017 and 2018 growing seasons.

Treatments *	TAM_0%_ (Control)		TAM_0.25%_		TAM_0.5%_		TAM_1%_	
Parameters	2017	2018	2017	2018	2017	2018	2017	2018
plant Height (cm)	67.33 ± 4.93 ^e^	70.00 ± 4.58 ^d^	81.00 ± 0.23 ^b^	80.33 ± 1.15 ^b^	88.00 ± 2.00 ^a^	80.33 ± 0.58 ^b^	72.67 ± 1.15 ^c^	74.00 ± 3.46 ^c^
Increase/Decrease Rate (%)	(00.00)	(00.00)	(20.30)	(14.76)	(30.70)	(14.76)	(7.93)	(5.71)
Branches (No.)	4.67 ± 0.58 ^b^	4.33 ± 0.58 ^b^	6.00 ± 0.65 ^a^	6.00 ± 0.53 ^a^	6.00 ± 0.80 ^a^	6.00 ± 0.83 ^a^	4.67 ± 0.58 ^b^	4.33 ± 0.58 ^b^
Increase/Decrease Rate (%)	(00.00)	(00.00)	(28.48)	(38.57)	(28.48)	(38.57)	(00.00)	(00.00)
Chlorophyll (mg 100 g^−1^ FW)	60.31 ± 2.05 ^a^	59.68 ± 1.67 ^c^	64.54 ± 2.44 ^a^	65.32 ± 2.09 ^b^	65.80 ± 5.03 ^a^	68.74 ± 0.81 ^a^	62.44 ± 4.83 ^a^	60.81 ± 1.07 ^c^
Increase/Decrease Rate (%)	(00.00)	(00.00)	(7.03)	(9.46)	(9.11)	(15.19)	(3.54)	(1.90)
Dry matter (%)	33.17 ± 1.73 ^c^	33.31 ± 0.87 ^b^	36.00 ± 1.52 ^ab^	35.90 ± 1.50 ^a^	36.80 ± 1.17 ^a^	36.54 ± 1.15 ^a^	34.11 ± 0.45 ^bc^	33.45 ± 0.88 ^b^
Increase/Decrease Rate (%)	(00.00)	(00.00)	(8.53)	(7.77)	(10.94)	(9.70)	(2.83)	(0.42)
Leaf-N (mg 100 g^−1^ DW)	1.32 ± 0.01 ^a^	1.35 ± 0.08 ^a^	1.19 ± 0.04 ^b^	1.22 ± 0.09 ^a^	1.21 ± 0.06 ^b^	1.17 ± 0.07 ^a^	1.16 ± 0.04 ^b^	1.23 ± 0.13 ^a^
Increase/Decrease Rate (%)	(00.00)	(00.00)	(−9.87)	(−9.63)	(−8.10)	(−13.33)	(−13.09)	(−9.13)
Leaf-P (mg 100 g^−1^ DW)	0.41 ± 0.01 ^b^	0.42 ± 0.03 ^b^	0.53 ± 0.03 ^a^	0.52 ± 0.02 ^a^	0.51 ± 0.03 ^a^	0.51 ± 0.02 ^a^	0.45 ± 0.04 ^b^	0.44 ± 0.01 ^b^
Increase/Decrease Rate (%)	(00.00)	(00.00)	(29.27)	(23.81)	(25.20)	(20.64)	(10.56)	(4.76)
Leaf-K (mg 100 g^−1^ DW)	3.97 ± 0.18 ^c^	4.13 ± 0.09 ^c^	5.49 ± 0.35 ^a^	5.21 ± 0.09 ^a^	5.26 ± 0.43 ^ab^	5.44 ± 0.21 ^a^	4.59 ± 0.44 ^bc^	4.49 ± 0.06 ^b^
Increase/Decrease Rate (%)	(00.00)	(00.00)	(38.29)	(26.07)	(32.41)	(31.72)	(15.62)	(8.64)

* Represented Data (*n* = 3) were mean ± SD. Different superscript letters in each row indicate significant differences (*p* < 0.05).

**Table 5 plants-10-01045-t005:** The effect of different TAM^®^ levels on fruit characterizations of *C. annuum,* during 2017 and 2018 growing seasons.

Treatments *	TAM_0%_ (Control)		TAM_0.25%_		TAM_0.5%_		TAM_1%_	
Parameters	2017	2018	2017	2018	2017	2018	2017	2018
Total Yield (kg m^2^)	3.27 ± 0.05 ^c^	3.12 ± 0.03 ^b^	3.75 ± 0.53 ^b^	3.76 ± 0.03 ^a^	4.32 ± 0.09 ^a^	3.58 ± 0.03 ^a^	3.5 ± 0.02 ^bc^	3.56 ± 0.33 ^a^
Increase/Decrease Rate (%)	(00.00)	(00.00)	(14.67)	(17.50)	(32.22)	(23.20)	(7.30)	(14.00)
Fruit Length (cm)	11.82 ± 0.32 ^b^	11.97 ± 0.07 ^a^	12.21 ± 025 ^ab^	12.19 ± 0.27 ^a^	12.60 ± 0.09 ^a^	15.68 ± 5.09 ^a^	11.88 ± 0.38 ^b^	12.03 ± 0.05 ^a^
Increase/Decrease Rate (%)	(00.00)	(00.00)	(3.36)	(1.87)	(6.63)	(31.06)	(0.56)	(0.56)
Fruit Diameter (cm)	1.16 ± 0.04 ^b^	1.11 ± 0.10 ^a^	1.23 ± 0.02 ^a^	1.17 ± 0.13 ^a^	1.24 ± 0.03 ^a^	1.27 ± 0.02 ^a^	1.17 ± 0.03 ^b^	1.11 ± 0.10 ^a^
Increase/Decrease Rate (%)	(00.00)	(00.00)	(6.03)	(5.41)	(6.90)	(14.41)	(0.86)	(0.00)

* Represented Data (*n* = 3) were mean ± SD. Different superscript letters in each row indicate significant differences (*p* < 0.05).

## Data Availability

Not applicable.

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
