# Peer review of "Impact of Commercial Seaweed Liquid Extract (TAM®) Biostimulant and Its Bioactive Molecules on Growth and Antioxidant Activities of Hot Pepper (Capsicum annuum)"

_plants, 2021, doi:10.3390/plants10061045_

Round 1
Reviewer 1 Report
Designing and creating new biostimulants (especially in foliar application) requires accurate testing of the product results on the plant’s morpho-physiological traits and a deep understanding of the mechanism of action of chosen products during various environmental conditions and developmental stages. 1. What is the plant’s morpho-physiological and mechanism of action of the chosen product?
2. What are the main physiological mechanisms related to the quantity-quality effects of new products on the crop?
3. Nutrient uptake data are needs when the research comparing conventional fertilizer and new fertilizer application.
4. Identification of the right time of biostimulant application is as important as the determination of the exact dose, in order to avoid waste of product, high production costs, and unexpected results. What is the best time application? early-stage growth or flowering or maturity? the research should support these gaps.
The paper and the abstract indicated:
''These findings indicate that the extract of sea-weed can be used as an environmentally friendly, multi-functional biostimulant, in the agricultural field for more sustainable production. The use of seaweed as a biostimulant could contribute significantly to alleviating the adverse effects of the global major nutrient deficiencies, vigorous biotic and abiotic stresses associated with climate change, in addition to reducing the use of hazardous synthetic fertilizers.''
1. How you conclude that the use of seaweed as a biostimulant could contribute significantly to alleviating the adverse effects of the vigorous biotic and abiotic stresses associated with climate change. Do you have research or data in the current experiment?
2.I suggest a major re-write of the introduction. It should provide an overview of the importance of the paper topic, problem statements, and hypothesis.
Author Response
SUMMARY OF AUTHOR(S) RESPONSE TO REVIEWER’S COMMENTS
Manuscript #: plants-1210925
Title of the Manuscript: Impact of Commercial Seaweed Liquid Extract (TAM®) Biostimulant and its Bioactive Molecules on Growth and Antioxidant Activities of Hot Pepper (Capsicum annuum)
Authors: Mohamed Ashour, Shimaa M. Hassan, Gamal A.G. Ammar, Ahmed Gaber, Walaa F. Alsanie, Rania El-Shenody, Abdallah Tageldein Mansour, Mostafa E. Elshobary
|
Comments of Reviewer 1# |
Author(s) response |
|
|
|
|
Designing and creating new biostimulants (especially in foliar application) requires accurate testing of the product results on the plant’s morpho-physiological traits and a deep understanding of the mechanism of action of chosen products during various environmental conditions and developmental stages. 1. What is the plant’s morpho-physiological and mechanism of action of the chosen product? |
The mode of action of the most natural biostimulants, including the commercial seaweed-based products, is not cleared until now because they contain a mixture of macro & microelements plus some bioactive compounds that work synergistically to improve plant growth and enhance plant content of ioactive molecules. During this study, plant height (cm), number of branches (n), leaves chlorophyll (mg 100 g fresh weight), dry matter (%), nitrogen (N), phosphorus (P), and potassium (K) were determined as morpho-physiological treats. As well as, the bioactive components and their applications including, ascorbic acid, total flavonoids compounds, total phenolic compounds, total antioxidant capacity, and DPPH inhibition % have been selected to distinguish among treatments. |
|
What are the main physiological mechanisms related to the quantity-quality effects of new products on the crop? |
Our paper has focused only on the effect of TAM® on morpho-agronomic and bioactive properties (mentioned in the previous response) of hot pepper, which have been improved by TAM® treatments. |
|
Nutrient uptake data are needs when the research comparing conventional fertilizer and new fertilizer application. |
We are sorry this data is not available. This suggestion will be taken in our consideration in fthe uture work. |
|
Identification of the right time of biostimulant application is as important as the determination of the exact dose, in order to avoid waste of product, high production costs, and unexpected results. What is the best time application? early-stage growth or flowering or maturity? the research should support these gaps. |
We appreciate the reviewer's suggestion. however application time was not a studied parameter in this study, we were only concerned that all growth stages receive equal amounts of water two times weekly during the growing trials till maturation. |
|
The paper and the abstract indicated: ''These findings indicate that the extract of sea-weed can be used as an environmentally friendly, multi-functional biostimulant, in the agricultural field for more sustainable production. The use of seaweed as a biostimulant could contribute significantly to alleviating the adverse effects of the global major nutrient deficiencies, vigorous biotic and abiotic stresses associated with climate change, in addition to reducing the use of hazardous synthetic fertilizers.'' 1. How you conclude that the use of seaweed as a biostimulant could contribute significantly to alleviating the adverse effects of the vigorous biotic and abiotic stresses associated with climate change. Do you have research or data in the current experiment? |
No, we don’t have this type of data accordingly, this statement has been modified to ''These findings indicate that the extract of seaweed can be used as an environmentally friendly, multi-functional biostimulant, in the agricultural field for more sustainable production, in addition to reducing the use of hazardous synthetic fertilizers.” (Page: 1, Lines: 38-40) |
|
2. Suggest a major re-write of the introduction. It should provide an overview of the importance of the paper topic, problem statements, and hypothesis. |
Introduction has been rewritten again to focus on the paper topic, problem statements, and hypothesis, according to the reviewer recommendation |
We would like to extend our sincere thanks and appreciation to Reviewer # 1 and the editorial board. In fact, their comments and guidance added a lot to the research and increased its scientific content. Therefore, the words cannot express their gratitude for their time and effort they put into evaluating this research.

Reviewer 2 Report
Manuscript plants-1210925 refers to the plant biostimulant effect of a seaweed extract (TAM®) on hot pepper (Capsicum annuum). The manuscript plants-1210925 share the same hypothesis and methods with the previous papers published by the same group of authors in MDPI journals in 2021. Experiments were done in the same period. The difference is related to type of vegetable on which the plant biostimulant was tested.
In the present form the manuscript is flawed by the extensive text recycling from the previous papers reporting the effects of the same seaweed extract (TAM®) on rocket salad (Hassan, S. M., Ashour, M., Soliman, A. A., Hassanien, H. A., Alsanie, W. F., Gaber, A., & Elshobary, M. E., 2021, The Potential of a New Commercial Seaweed Extract in Stimulating Morpho-Agronomic and Bioactive Properties of Eruca vesicaria (L.) Cav. Sustainability, 13(8), 4485) and cucumber (Hassan, S. M., Ashour, M., Sakai, N., Zhang, L., Hassanien, H. A., Gaber, A., & Ammar, G. , 2021, Impact of Seaweed Liquid Extract Biostimulant on Growth, Yield, and Chemical Composition of Cucumber (Cucumis sativus). Agriculture, 11(4), 320).
Partial text recycling is acceptable for the non-native English speakers in Material and Method Section and in the Introduction Section referring to hypothesis. However, the text recycling is done in a superficial manner. Lines L181 and L188 from manuscript plants-1210925, referring to hot pepper (Capsicum annuum)are “2.3.2. Cucumis sativus Yield and Fruit Parameters” and “2.3.3. Antioxidant Activities of TAM® and C. sativus Fruits “. Such a superficial text recycling, for a text written and verified by 4 authors (according to Contribution of the authors Section) is not acceptable.
Recycling of the text is obvious also in the Discussion Section. L394-L398, manuscript plants-1210925: “Our findings were consistent with those of Hamed et al. [16] who reported that seaweed extracts could contain several nutrients, essential minerals, vitamins, phytohormones, ascorbic acids, and a variety of other bioactive components [5,6]”. Hassan et al. 2021, Agriculture, 11(4), 320: “Our findings were in line with those of Hamed et al. [55], who stated that seaweed extracts could contain nutrients, trace minerals, vitamins, phytohormones, ascorbic acids, and many other bioactive compounds [56].”
Another issue is related to consideration of the seaweed extract as a replacer for fertilizers. L129-L134 “The current experimental trial was conducted to investigate the impact of 50% partial replacement of traditional NPK mineral fertilization by three different levels (2.5 ppt, 5 ppt, and 10 ppt) of seaweed foliar spray TAM® (TAM2.5, TAM5, and TAM10, respectively)….” By definition plant biostimulants (including seaweed extracts) are not fertilizer. Plant biostimulant enhance / benefits nutrient uptake - Du Jardin, Patrick. "Plant biostimulants: definition, concept, main categories and regulation." Scientia Horticulturae 196 (2015): 3-14.
Such formulations could mislead the reader and need to be carefully re-written.
Authors use in a wrong way ppt concentration. By definition ppt is “part per trillion”, respectively 10-12. The concentration of their seaweed extract is in the range of 0,25-1%, i.e., 10-3 – 10-2. Authors confuse concentrations of the applied foliar spray, with the dose, which is the quantity of the seaweed extract applied per surface unit. Information regarding spraying volume applied per m2 or ha (or even feddan) is not provided – therefore their work is not reproducible by the others.
Figure 1 is followed by Figure 3 – Figure 2 is missing. Error bars are present in these two figures – information regarding type of error bars and their significance is missing.
Authors use feddan / fadden as a surface unit. In a scientific paper units outside of the SI system should not be use. According to Instruction for authors from Plants website, https://www.mdpi.com/journal/plants/instructions
“SI Units (International System of Units) should be used. Imperial, US customary and other units should be converted to SI units whenever possible.” Conversion to fadden to m2 (or at the limit with SI tolerated unit, hectare) must be done.
Manuscript must be carefully re-written before being re-considered for publication. English must be improved. A clearer presentation of the present manuscript with the previous published papers related to effects Mistakes such as keeping text from previous papers not-related to the present manuscript and figure numbering should be corrected. Essential information related to reproduction of the experiments by other scientists must be added.
Author Response
SUMMARY OF AUTHOR(S) RESPONSE TO REVIEWER’S COMMENTS
Manuscript #: plants-1210925
Title of the Manuscript: Impact of Commercial Seaweed Liquid Extract (TAM®) Biostimulant and its Bioactive Molecules on Growth and Antioxidant Activities of Hot Pepper (Capsicum annuum)
Authors: Mohamed Ashour, Shimaa M. Hassan, Gamal A.G. Ammar, Ahmed Gaber, Walaa F. Alsanie, Rania El-Shenody, Abdallah Tageldein Mansour, Mostafa E. Elshobary
|
Comments of Reviewer 2 # |
Author(s) response |
|
Manuscript plants-1210925 refers to the plant biostimulant effect of a seaweed extract (TAM®) on hot pepper (Capsicum annuum). The manuscript plants-1210925 share the same hypothesis and methods with the previous papers published by the same group of authors in MDPI journals in 2021. Experiments were done in the same period. The difference is related to type of vegetable on which the plant biostimulant was tested. |
|
|
In the present form the manuscript is flawed by the extensive text recycling from the previous papers reporting the effects of the same seaweed extract (TAM®) on rocket salad (Hassan, S. M., Ashour, M., Soliman, A. A., Hassanien, H. A., Alsanie, W. F., Gaber, A., & Elshobary, M. E., 2021, The Potential of a New Commercial Seaweed Extract in Stimulating Morpho-Agronomic and Bioactive Properties of Eruca vesicaria (L.) Cav. Sustainability, 13(8), 4485) and cucumber (Hassan, S. M., Ashour, M., Sakai, N., Zhang, L., Hassanien, H. A., Gaber, A., & Ammar, G. , 2021, Impact of Seaweed Liquid Extract Biostimulant on Growth, Yield, and Chemical Composition of Cucumber (Cucumis sativus). Agriculture, 11(4), 320). Partial text recycling is acceptable for the non-native English speakers in Material and Method Section and in the Introduction Section referring to hypothesis. However, the text recycling is done in a superficial manner. Lines L181 and L188 from manuscript plants-1210925, referring to hot pepper (Capsicum annuum) are “2.3.2. Cucumis sativus Yield and Fruit Parameters” and “ 2.3.3. Antioxidant Activities of TAM® and C. sativus Fruits “. Such a superficial text recycling, for a text written and verified by 4 authors (according to Contribution of the authors Section) is not acceptable. |
We are sorry for these mistakes. The whole manuscript has been revised according to reviewer comments |
|
Recycling of the text is obvious also in the Discussion Section. L394-L398, manuscript plants-1210925: “Our findings were consistent with those of Hamed et al. [16] who reported that seaweed extracts could contain several nutrients, essential minerals, vitamins, phytohormones, ascorbic acids, and a variety of other bioactive components [5,6]”. Hassan et al. 2021, Agriculture, 11(4), 320: “Our findings were in line with those of Hamed et al. [55], who stated that seaweed extracts could contain nutrients, trace minerals, vitamins, phytohormones, ascorbic acids, and many other bioactive compounds [56]. |
This sentence has been modified |
|
Another issue is related to consideration of the seaweed extract as a replacer for fertilizers. L129-L134 “The current experimental trial was conducted to investigate the impact of 50% partial replacement of traditional NPK mineral fertilization by three different levels (2.5 ppt, 5 ppt, and 10 ppt) of seaweed foliar spray TAM® (TAM2.5, TAM5, and TAM10, respectively)….” By definition plant biostimulants (including seaweed extracts) are not fertilizer. Plant biostimulant enhance / benefits nutrient uptake - Du Jardin, Patrick. "Plant biostimulants: definition, concept, main categories and regulation." Scientia Horticulturae 196 (2015): 3-14. Such formulations could mislead the reader and need to be carefully re-written. |
We're not talking about seaweed extract as an alternative to fertilizer, but we're talking about using seaweed extract as biostimulator to reduce the chemical fertilizer utilization by 50 %, perhaps by the role of biostimulant to increase nutrient absorption efficiency. This sentence has been revised more clearly. |
|
Authors use in a wrong way ppt concentration. By definition ppt is “part per trillion”, respectively 10-12. The concentration of their seaweed extract is in the range of 0,25-1%, i.e., 10-3 – 10-2. Authors confuse concentrations of the applied foliar spray, with the dose, which is the quantity of the seaweed extract applied per surface unit. Information regarding spraying volume applied per m2 or ha (or even feddan) is not provided – therefore their work is not reproducible by the others. |
We are sorry for this mistake and the concentrations have been modified to percent throughout the manuscript. the volume of foliar spray was 100-200 ml/m2 which is mentioned in material and method section. |
|
Figure 1 is followed by Figure 3 – Figure 2 is missing. Error bars are present in these two figures – information regarding type of error bars and their significance is missing. |
The figures number have been rechecked. error bars and significant information have been added |
|
Authors use feddan / fadden as a surface unit. In a scientific paper units outside of the SI system should not be use. According to Instruction for authors from Plants website, https://www.mdpi.com/journal/plants/instructions “SI Units (International System of Units) should be used. Imperial, US customary and other units should be converted to SI units whenever possible.” Conversion to fadden to m2 (or at the limit with SI tolerated unit, hectare) must be done. |
This unit has been modified to m2 |
|
Manuscript must be carefully re-written before being re-considered for publication. English must be improved. A clearer presentation of the present manuscript with the previous published papers related to effects Mistakes such as keeping text from previous papers not-related to the present manuscript and figure numbering should be corrected. Essential information related to reproduction of the experiments by other scientists must be added. |
The manuscript has been carefully reviewed and all mentioned issues have been considered. |
We would like to extend our sincere thanks and appreciation to the reviewers and editorial board. In fact, their comments and guidance added a lot to the research and increased its scientific content. Therefore, the words cannot express their gratitude for their time and effort they put in evaluating this research.

Reviewer 3 Report
The work titled "Impact of Commercial Seaweed Liquid Extract (TAM®) Biostimulant and its Bioactive Molecules on Growth and Antioxidant Activities of Hot Pepper (Capsicum annuum)" is about the effect of the foliar application of different concentrations of a patented biostimulant on hot pepper growth, yield and its fruit nutritional and antioxidant properties. Its usage as a 50% substitute of traditional NPK fertilization is sounded, since it perfectly fits the application purposes of PLANTS.
However, I suggest to improve some article sections.
Materials and Methods:
I suggest to add some information about the environment conditions during the experiment along the two seasons. Also, was the experiment in pots and was it on the same plants along the two seasons?
Please check the biostimulant provived doses in Paragraph 2.2.2. The ml/L quantity are appropriated, but their conversion into ppt unit seems not correct.
Please change Cucumis sativus into Capsicum annuum in paragraph 2.3.2 and 2.3.3.
Results:
Figure 3 requires the appropriate statystical reference letters and analyses.
Abstract:
Line 36: The highest TAM concentration is not the one which provides the highest yield. The best TA concentration is instead 5 ml/L as reported in the conclusion section.
English language requires some adjustments.
For example:
Line 66: change "improving" into "they improve"
Line 176: change "leave's" into "leaf"
Line 212: change "the higest significant" into "higher"
Line 213: change "the lowest significant" into "lower"
Line 241: change "the the higher significance in" into "higher"
Author Response
SUMMARY OF AUTHOR(S) RESPONSE TO REVIEWER’S COMMENTS
Manuscript #: plants-1210925
Title of the Manuscript: Impact of Commercial Seaweed Liquid Extract (TAM®) Biostimulant and its Bioactive Molecules on Growth and Antioxidant Activities of Hot Pepper (Capsicum annuum)
Authors: Mohamed Ashour, Shimaa M. Hassan, Gamal A.G. Ammar, Ahmed Gaber, Walaa F. Alsanie, Rania El-Shenody, Abdallah Tageldein Mansour, Mostafa E. Elshobary
|
Comments of Reviewer 3 # |
Author(s) response |
|
The work titled "Impact of Commercial Seaweed Liquid Extract (TAM®) Biostimulant and its Bioactive Molecules on Growth and Antioxidant Activities of Hot Pepper (Capsicum annuum)" is about the effect of the foliar application of different concentrations of a patented biostimulant on hot pepper growth, yield and its fruit nutritional and antioxidant properties. Its usage as a 50% substitute of traditional NPK fertilization is sounded, since it perfectly fits the application purposes of PLANTS. However, I suggest to improve some article sections. Materials and Methods:
|
|
|
I suggest to add some information about the environment conditions during the experiment along the two seasons. Also, was the experiment in pots and was it on the same plants along the two seasons? |
The planting system that was adopted in the current study, by planting the hot pepper seedlings under greenhouse aimed at providing a cover to protect the plants from extreme conditions outside the green-house like, wind and dust. In addition, as the green-house maintains a slightly higher temperature inside than the normal temperature outside, this slight difference (few degrees) would protect the plants from the normal very low night temperatures at seedling establishment, during February. However, similar to the majority of the Mediterranean countries, we do not control the temperature and the humidity inside the green-house (no need for that), we just regularly measure them but we didn't measure the light intensity. Information about the average monthly temperature and humidity measured inside the green-house is added to the Materials and Methods section (Page: 3, Line 144–147) |
|
Please check the biostimulant provived doses in Paragraph 2.2.2. The ml/L quantity are appropriated, but their conversion into ppt unit seems not correct. |
This mistake has been corrected by converted to (%) |
|
Please change Cucumis sativus into Capsicum annuum in paragraph 2.3.2 and 2.3.3. |
We so sorry for this mistake. This mistake has been corrected |
|
Results: |
|
|
Figure 3 requires the appropriate statistical reference letters and analyses. |
Done, and the number of Fig. has been corrected to Fig. 2. |
|
Abstract: |
|
|
Line 36: The highest TAM concentration is not the one which provides the highest yield. The best TA concentration is instead 5 ml/L as reported in the conclusion section. |
It has been changed to TAM0.5% (Page:1, Line: 34). |
|
English language requires some adjustments, For example: Line 66: change "improving" into "they improve" |
Corrected (Page: 2, Line: 56) |
|
Line 176: change "leave's" into "leaf" |
Corrected (Page: 6, Line: 177) |
|
Line 212: change "the higest significant" into "higher" |
Corrected (Page: 7, Line: 214) |
|
Line 213: change "the lowest significant" into "lower" |
Corrected (Page: 7, Line: 215) |
|
Line 241: change "the the higher significance in" into "higher" |
Corrected (Page: 8, Line: 244 ) |
We would like to extend our sincere thanks and appreciation to the reviewers and editorial board. In fact, their comments and guidance added a lot to the research and increased its scientific content. Therefore, the words cannot express their gratitude for their time and effort they put in evaluating this research.

Round 2
Reviewer 1 Report
Accept in present form
Reviewer 2 Report
Authors made the requested improvements. Manuscript still have a low originality, being the last from a line of papers of the same groups presenting the biostimulant effects of seaweed extract on various vegetables species. Extensive english editing is required.